# Emotional–Behavioral Disorders in Healthy Siblings of Children with Neurodevelopmental Disorders

**DOI:** 10.3390/medicina56100491

**Published:** 2020-09-23

**Authors:** Mariarosaria Caliendo, Valentina Lanzara, Luigi Vetri, Michele Roccella, Rosa Marotta, Marco Carotenuto, Daniela Russo, Francesco Cerroni, Francesco Precenzano

**Affiliations:** 1Centro di Riabilitazione “La Filanda-Lars”, 84087 Sarno, SA, Italy; m.caliendo@hotmail.it (M.C.); daniela.1991@hotmail.it (D.R.); 2Clinic of Child and Adolescent Neuropsychiatry, Università degli Studi della Campania “Luigi Vanvitelli”, 80138 Naples, Italy; valentina.lanzara@unicampania.it (V.L.); marco.carotenuto@unicampania.it (M.C.); cerronifrancesco@gmail.com (F.C.); f.precenzano@hotmail.it (F.P.); 3Department of Sciences for Health Promotion and Mother and Child Care “G. D’Alessandro”, University of Palermo, via del Vespro 129, 90127 Palermo, Italy; 4Oasi Research Institute-IRCCS, 94018 Troina, Italy; 5Department of Psychology, Educational Science and Human Movement, University of Palermo, 90128 Palermo, Italy; michele.roccella@unipa.it; 6Department of Health Sciences, University “Magna Graecia”, 88100 Catanzaro, Italy; marotta@unicz.it

**Keywords:** autism spectrum disorders, Down’s syndrome, SDQ, siblings, fraternal relationship, emotional disorder, behavioral disorders

## Abstract

*Background and Objectives:* Siblings of disabled children are more at risk of developing mental illnesses. More than 50 international studies show that about 8% of children and adolescents suffer from a mental disorder, which is almost always a source of difficulties both at the interpersonal level (in the family and with peers) and at school. Healthy siblings of children with disabilities are one of the groups most at risk for consequences in psychological health and well-being. As some authors suggest, siblings build their idea of “being people”, in terms of character and personality, by continuously and daily confronting themselves with the theme of disability and a family context subjected to continuous stress. The following contribution aims to compare emotional–behavioral disorders in healthy siblings of children with autism spectrum disorder, in healthy siblings of children with Down’s syndrome and in healthy siblings of children with typical development. *Materials and Methods*: The results involve 153 children from the region of Campania and their caregivers through the administration of the Strength and Difficulties Questionnaire. *Results*: From the data, it emerged that siblings of children with autism spectrum disorder and siblings of children with Down’s syndrome have a greater emotional fragility, especially among male subjects. *Conclusions:* Our results require us to reflect on the clinical and policy measures needed to ensure the well-being of siblings of disabled children, mainly through appropriate sibling coping training.

## 1. Introduction

Having a disabled person within the household leads to profound changes. Since the family functions as a system and as an emotional unit, it is not possible to consider the components as single units. On the contrary, it seems more appropriate to care about the individuals belonging to the same system [1].

International studies showed that about 8% of children and adolescents suffer from a mental disorder, which is almost always a source of difficulty both at the interpersonal level (in the family and with peers) and at school [2].

The most recurring question asked by parents of a disabled child, whether the disability is from birth or has developed later, is why it happened. It is often thought that the question exclusively belongs to parents’ reflections, however, as several studies highlight [3], this question is also shared by other children, regardless of gender and order of birth.

The birth of a disabled child is a critical event for all members of the family, and, in some cases, it can be very difficult to manage. For this reason, it is essential to take into account the development stages the siblings have to go through, considering that the influence of the disability can be different. Having a fraternal relationship with a person who has physical, emotional, relational and/or cognitive difficulties is a challenge which cannot be avoided by siblings and it encompasses a very long period of their lives. The fraternal bond, together with other significant relationships, contributes to the formation of identity, so the condition of healthy siblings is a different experience, which can lead to different outcomes in the psychological and sociological well-being of the subject.

The number of studies on siblings with emotional–behavioral disorders has increased in recent decades, although most of them concern children with cognitive disabilities [4]. However, the results obtained cannot be generalized, because the demands of different types of medical condition and the adaptability of their families are slightly different [5]. Therefore, these studies are not applicable to children with physical disabilities or chronic diseases.

Today, it is known that growing up with a sibling with a disability or chronic illness has both positive and negative effects [6]. However, while the negative effects are rather immediate, the positive effects are found later, in adolescence or even in adulthood [7]. Most siblings of children with disabilities or chronic diseases cope well with the situation [8], while a minority risks developing severe adaptation difficulties [4,6,7], such as school problems, decreased self-esteem and social stigma [9]. The presence of a person with disabilities creates many imbalances within the family. Indeed, being a parent to a child with disabilities can sometimes be an emotionally devastating experience, which can also be true for the siblings.

As A. Dondi properly claims, siblings build their idea of “being people”, in terms of character and personality, by continuously and daily confronting themselves with the theme of disability and a family context subjected to continuous stress [10]. This study stems from the intention of identifying the existing or potential problems of siblings of individuals with chronic disabilities, in a special care path.

As the international literature shows, a large percentage of children and adolescents suffer from a mental disorder, this is a source of difficulty at school and in the relationship with peers. Healthy siblings of people with disabilities are the group most at risk. In one of his studies, Kaminsky reported fewer prosocial behaviors in the siblings of autistic children, as well as less competition and fewer quarrels in their fraternal relationships than in the control group [11]. Similarly, De Caroli and Sagone, in their study, analyzed the social attitudes of unaffected siblings toward disabled brothers and sisters, showing more negative social attitudes and more negative representations of their autistic brothers compared to siblings of brothers/sisters with Down’s syndrome or intellectual disability [12]. Another study, conducted on 30 pairs of children with autism spectrum disorder (ASD) or Down’s syndrome (DS), compared the interactions of autistic children with their siblings and those of Down’s children with their siblings. It found that autistic children, spending less time with their siblings, use a smaller repertoire of prosocial [13] and antagonistic initiatives towards their siblings and imitate them less [14].

The different results found in the literature can be partially explained by the different hypotheses or by some methodological problems such as the lack of an appropriate control group, indirect measures provided by parents and teachers and the retrospective nature of some studies. Despite the variability of these results, most of the empirical evidence gathered suggests that a disabled child in a family has a significant influence on the psychosocial development of healthy children.

There are very few Italian studies investigating the neuropsychological implications in preadolescent siblings of disabled people. Nevertheless, there is much Italian evidence about adult siblings of people with intellectual or developmental disabilities, showing higher levels of depression and anxiety, lower levels of life satisfaction and closeness and worries [15,16].

However, there are literature data indicating that younger siblings have been reported to engage more frequently in negative behaviors with their siblings with special needs [17].

A recent Italian study assessed the psychological adjustment of 26 normally developing siblings of high-functioning children with ASD, finding that a minority of cases are at risk of social impairments (7.7%), internalizing (23.1%), externalizing (3.8%) and total difficulties (11.5%) and of distress in the parent–child system (15%) [18].

This work is based on a study carried out in Iran in 2016 on 174 children [19]. This study showed that the emotional–behavioral disorders of siblings in the families of children with autism were more severe than those of the two comparison groups. Additionally, both siblings of children with autism and siblings of children with Down’s syndrome have a greater admiration for their siblings and fewer quarrels and less competition in their relationships than children in the control group with typical development. Despite problems such as depression, anxiety, hyperactivity and difficulty in communicating with peers, having siblings with autism can probably increase prosocial behaviors [14].

The data reported above cannot be generalized to the Italian territory.

The same type of sample used in our study has also been chosen by an Iranian study. That is, aged between 3 and 9 years and two groups of diseases: autism and Down’s syndrome, which are, to date, the most frequent groups of diseases.

The first group of children examined is affected by ASD. It should be noted that the term ASD refers to a neurodevelopmental disorder, biologically determined and with an onset in the first three years of life, leading to a permanent disability that mainly affects the areas of mutual social interaction, the ability to communicate ideas and feelings and the ability to establish relationships with others and, often, with a specific profile of cognitive weaknesses [19,20,21,22]. The other group of children, on the other hand, includes healthy siblings of children with Down’s syndrome or a genetic disease (genomic mutation), the first genetic cause of intellectual disability, in which chromosome 21 is supernumerary.

## 2. Materials and Methods

The following study was carried out in Campania in 2018 on 159 healthy children divided into three groups aged between 3 and 9 years, of which 53 cases had siblings with typical development, 53 cases had siblings with Down’s syndrome and 53 cases had siblings with autism.

This research had approval from Ethical Committee (Prot. n. 13883; EuDract 2015-001159-66, date: 9 March 2015).

The participants were selected according to the convenience sampling method and recruited partly from rehabilitation centers in the province of Salerno, from the Rehabilitative Unit “La Filanda-Lars”, Sarno (SA), Italy and partly from a nursery and primary school in the province of Salerno. The sample of subjects chosen had an age range of 3 to 9 years, the presence of a subject with autism or Down’s syndrome, with a diagnosis at least six months previously, and families with at least two children.

Children with autism had a score from 33 to 60 on the Childhood Autism Rating Scale test (CARS2) by Eric Schopler et al., a tool specifically developed to identify children with autism from two years of age, divided into fifteen items relating to the main areas of behavior to which one can give a score from 1 to 4. The sum of all scores gives an overall value with the following meanings: from 15 to 30: non-autistic; from 30 to 37: light to medium autism; from 37 to 60: severe autism. Intellectual disability or borderline intellectual functioning were exclusion criteria in order to avoid emotional awareness impairment [23,24].

Caregivers were asked to fill in the Strength and Difficulties Questionnaire (SDQ-ITA) and to provide the following information: pathology of the child, sex, age, number of children and who completed the questionnaire. The SDQ-ITA questionnaire on strengths and weaknesses is a short and easy to administer tool developed by Goodman (1967) in Great Britain which follows the structure of the items of the Rutter tool. It is a concise instrument that allows for obtaining a lot of information on the behavior of the child. The test contains 25 items referring to positive or negative attributes of the child’s behavior.

The items can be divided into five subscales:Hyperactivity, which also contains items on attention problems;Behavioral problems;Emotional difficulties, especially aspects related to anxiety and depression;Prosocial behavior;Relationships with peers [25,26,27].

The evaluator has a three-point Likert scale to indicate how much a certain attribute is descriptive of the child’s behavior. The ratings range from 0 (not true) to 1 (partially true) to 2 (absolutely true). Five items describe positive behavior (items 7, 11, 14, 21 and 2), so it is necessary to reverse the score before adding them to the scores of the five subscales. On the scoring sheet, the reversed items are easily recognizable, as 0 corresponds to “absolutely true”, 2 to “not true” and 1 to “partially true”.

For all subscales, a higher score corresponds to a higher level of discomfort, except for the subscale of prosocial behavior, where a higher score indicates a significant presence of positive behaviors (Table 1) [28].

A value between the 80th and the 90th percentile indicates a slight psychological problem (subclinical level), a score higher than the 90th percentile indicates a more serious psychological problem (clinical level).

The data analysis was performed using IBM SPSS statistics software. First, descriptive statistics through chi-squared tests were computed for demographic data. Second, group comparisons were made using one-way analysis of variance (ANOVA). Statistical significance was set at *p* < 0.05.

## 3. Results

The total number of subjects participating in the study was 159, of which 58 were siblings of subjects diagnosed with autism, 58 had siblings diagnosed with Down’s syndrome and 58 had siblings with typical development.

Gender distribution is characterized by the presence of 29 males and 24 females in the autism group; 28 males and 25 females in the Down’s syndrome group; 24 males and 29 females in the control group (Table 2).

The questionnaire could be completed by the mother, the father or by another caregiver, a person who knew the child well. In all three groups, it was found that in most cases the mother completed the questionnaire with a percentage of 92.5% in the ASD and Down’s syndrome groups and a percentage of 84.9% in the control group. The fathers who completed the questionnaire made up 7.5% in the ASD group, 5.7% in the group of subjects with Down’s syndrome and 15.1% in the control group.

The mean age of the three groups was: 5.49 ± 1.95 for the siblings of autistic children, 5.94 ± 1.98 for the siblings of children with Down’s syndrome and 5.89 ± 1.98 for the control group.

The analysis of the number of siblings showed that the means were 2.34 ± 0.78 in the group of children with autism, 2.40 ± 0.71 for the group of subjects with Down’s syndrome and 2.26 ± 0.48 for the control group. The mode was two siblings for the ASD group (77.4%), in the group of subjects with Down’s syndrome (67.9%) and for the control group (75.5%) (Table 3).

In families with members with Down’s syndrome, it can be observed that there is a greater percentage of three children (28.3%) compared to families with an autistic subject, where the percentage of three children falls to 17.0%. Families with four children are at 1.9% for all three groups. Larger families with six children are only found in families with an autistic or Down’s syndrome subject.

The overall difficulties score of the SDQ was 19.92 ± 5.84 in the autism group, 19.75 ± 6.88 in the Down’s syndrome group and 14.67 ± 4.31 in the healthy group, highlighting a significantly more severe impairment in ASD and Down’s syndrome groups compared to the healthy group (*p* < 0.001).

In the analysis of the data, a single subscale was studied for each group.

Starting from the scale of emotionality (remember that a subclinical score is 4, while the clinical score is 6), it was found that 13.3% of subjects have clinical discomfort and 25.9% subclinical discomfort. Specifically, in the control group, the clinical discomfort is 0% while the subclinical discomfort is 24.5%. In the group of siblings of subjects with Down’s syndrome, clinical discomfort emerged in 11.5% of cases and subclinical discomfort in 30.8% of cases, while in the group of children with siblings with ASD, we note that 28.3% of subjects have clinical discomfort and 30.8% subclinical discomfort (Table 4).

The gender analysis showed that in the group of siblings of children with Down’s syndrome, females with clinical discomfort make up 8.3%, while males are at 14.3%, and in the group of siblings of autistic children, the clinical discomfort is found in 27.6% of males and 29.2% of females.

In the case of emotionality, the gender difference is statistically significant in males (*p* = 0.023) and females (*p* = 0.019).

For the scale related to behavioral problems (remember that a subclinical score is 4, while the clinical score is 5) it was found that 13.8% of subjects have clinical discomfort and 15.1% subclinical discomfort. Specifically, in the control group, the clinical discomfort is equal to 9.4%, while the subclinical discomfort is equal to 22.6%; in the group of siblings of subjects with Down’s syndrome, clinical discomfort emerged in 18.9% of cases and subclinical discomfort in 15.1% of cases, while in the group of siblings with ASD, we note that 13.2% of subjects have clinical discomfort and 7.5% subclinical discomfort (Table 5). These data show that there are no statistically significant differences between the control group and the siblings of subjects with autism or Down’s syndrome (*p* = 0.166). The gender difference is statistically significant.

For the scale relating to the problems of hyperactivity and inattention (remember that the subclinical cut-off is 6 and the clinical cut-off is 8), it was found that 9.4% of subjects have clinical discomfort and 15.7% subclinical discomfort. In the control group, the clinical discomfort is equal to 1.9%, while the subclinical discomfort is equal to 11.3%. In the group of siblings of subjects with Down’s syndrome, clinical discomfort emerged in 15.1% of cases and subclinical discomfort in 15.1% of cases; in the group of siblings with ASD, we note that 11.3% of subjects have clinical discomfort, while 20.8% subclinical discomfort (Table 6). From these data, we can deduce that there is a statistically significant difference between the control group and the brothers and sisters of subjects with autism or Down’s syndrome; on the contrary, there is no statistically significant difference between subjects with autism and Down’s syndrome (*p* = 0.083).

The gender analysis showed that in the group of siblings of children with Down’s syndrome, females have a prevalence of clinical discomfort of 8.3%, while for males it amounts to 21.4%; in the group of siblings of autistic children, clinical discomfort is found in 13.8% of males and 8.3% of females, while the control group shows that 0% of males have clinical discomfort, while females have a prevalence of clinical discomfort of 3.4%. The gender difference is statistically significant.

The scale of the relationship with peers (remember that a subclinical score is 3, while the clinical score is 5) showed that 20.1% of subjects have clinical discomfort and 23.3% subclinical discomfort. Specifically, in the control group, the prevalence of clinical discomfort is equal to 7.5% while the prevalence of subclinical discomfort is equal to 20.8%; in the group of siblings of subjects with Down’s syndrome, clinical discomfort emerged in 26.4% of cases and subclinical discomfort in 28.3% of cases, while in the group of children with siblings with ASD, we note that 26.4% of subjects have clinical discomfort and 20.8% subclinical discomfort (Table 7).

The gender difference, in this case, is not statistically significant.

The scale of prosociality (remember that the score is reversed so the subclinical cut-off is 4, while the clinical cut-off is 3) showed that 9.4% of subjects have clinical discomfort and 11.9% subclinical discomfort. Specifically, in the control group, the prevalence of clinical discomfort is equal to 7.5%, while the prevalence of subclinical discomfort is equal to 7.5%. In the group of siblings of subjects with Down’s syndrome, clinical discomfort emerged in 13.2% of cases and subclinical discomfort in 13.2% of cases; in the group of children with siblings with ASD, we note that 7.5% of subjects have clinical discomfort, while 15.1% have subclinical discomfort (Table 8). The siblings of subjects with autism and Down’s syndrome did not have significantly (*p* = 0.555) higher discomfort in prosocial behavior than in the control group. Gender analysis also showed that the difference is not statistically significant, for example, in the group of siblings of children with autism, females have a prevalence of clinical discomfort of 12.5%, while for males of 3.4% (*p* = 0.08 in males and *p* = 0.983 in females). In the group of siblings of children with Down’s syndrome, clinical discomfort is found in 17.9% of males and 8.0% of females, while in the control group, 4.2% of males and 10.3% of females have clinical discomfort.

## 4. Discussion

The study compared emotional and behavioral problems in siblings of children with autism, children with Down’s syndrome and children with typical development, involving 159 subjects.

The present study showed that siblings of autistic children and Down’s syndrome children (28.3% and 11.5%, respectively) are more at risk of emotional problems than those of the control group, where a percentage of 0% is found. Our results are in line with Pourbagheri et al.’s results and confirm, also in Italian siblings, that there was a significant difference in emotional–behavioral disorders among autism, Down’s syndrome, and healthy sibling groups. However, unlike Pourbagheri et al.’s studies, we did not find statistically significant differences related to the age of the subjects who were subject to the questionnaire, but we found statistically significant gender differences (see below).

This is due to a peculiar family environment that very often puts siblings in second place to the sibling with disabilities and causes a psychological distress condition. This could be somatically externalized, so these children would tend to complain more about headaches, stomachaches or nausea. They are also more worried, nervous or uncomfortable in new situations than the control group. The siblings of children with autism and Down’s syndrome have greater problems with attention and hyperactivity (respectively, 11.3% and 15.1%) than other peers, where the percentage is 1.9%. Siblings are more impulsive, restless and unable to stand still for a long time. There is a greater difficulty in concentrating, completing tasks and stopping to think before doing anything. From the data that emerged, the siblings of autistic children and Down’s syndrome children show greater peer relationship difficulties (26.4%) compared to the control group, where we found clinical discomfort in only 7.5% of cases. The lack of self-esteem would lead them to isolate themselves, to establish few relationships with peers and to prefer the company of adults. On the contrary, there was no statistically significant difference in the subscales of problems of conduct and prosociality compared to the control group. Therefore, despite the difficulties mentioned above, siblings have no problem sharing sweets, toys and pencils with other children. They often volunteer to help others and they generally comply with adult requests and rules.

In our study, we have chosen a relatively wide age range, varying from 3 to 9 years of age, in order to investigate preadolescent siblings because literature studies to date have mainly examined adult siblings of disabled individuals. The analysis, carried out with ANOVA, did not reveal a statistically significant difference related to the age of the subjects who were subject to the questionnaire. The same average between the group of siblings of children with autism and children with Down’s syndrome and the control group is shown. However, since some behavioral disorders can depend on the different development cycle of siblings, we hope that further studies could investigate separately preschool children and school-age children.

On the contrary, in this study, the gender difference was statistically significant, in particular for the scales of emotional distress, conduct and hyperactivity: males seem to have more emotional and behavioral problems than females. This is in line with other previous studies; in particular, groups of siblings of children with autism and Down’s syndrome have shown greater behavioral problems such as violence, aggression, poor social interaction with peers and greater feelings of competition; instead, the sisters have more anxiety, depression and low self-confidence [29]. This differential data can be linked to the greater ability to adapt and to positively cope with traumatic events in life, the so-called resilience, that is present more in women than in men. Moreover, women seem to have a higher quality of relationship with disabled sibs compared to men [30].

This disparity could also be confirmed in adulthood, as it can be said that fathers face their children’s disability differently than mothers do. In this regard, a study has shown that fathers’ expectations are more difficult to meet than those of mothers and that fathers are not very concerned with the daily activities and care of their children [31]. To support this, the results of the present study confirm that only a small percentage of the questionnaires (5.7% of the autism group, 7.5% of the Down’s group and 15.1% of the control group) were filled in by the fathers.

Therefore, we note that the results are not in line with the study that was carried out in Iran, where the gender difference was not statistically significant, as opposed to age.

## 5. Conclusions

In conclusion, in line with the above-presented results, the presence of children with autism and Down’s syndrome could cause greater psychological stress to their families and to their siblings. Despite the high emotional vulnerability and behavioral disorders of some of the siblings, the presence of a sibling with disabilities not only increases regulation, but it also improves exposure to adversity and disability. In fact, as has already been shown, growing up with a sibling with a disability can be both positive and negative for healthy siblings [32].

The family context is fundamental to the harmonious development of the subject, in fact, some siblings are placed in overprotective families with the disabled child who tend to exclude the other child; on the contrary, in other contexts, the healthy child is subjected to excessive demands on performance and to the burden of responsibility of the disabled sibling.

It is therefore essential to create protected places and moments, for the basic needs of siblings and, according to O’Shea et al. [33], there are four macro areas:
Receiving attention, to be considered within the care process but also in a context of family cohesion;Desiring to know, to be informed about the change that is necessarily altering the balance of the family system;Desiring to help, being able to play some role in the care of the sibling, alleviating the parental pressures regarding the condition of the illness;Desiring to have a routine like peers.

It would also be desirable to take into account, in taking charge of the patient, this component through the implementation of support groups for siblings, so that these children can live in a positive way and have an enriching experience regarding the disability of their sibling, without any consequences for their mental and psychological status. We believe that our results have important clinical and policy implications. Today, the main aid measures in Italy for families with a disabled child consist of material resources. However, the adopted measures are not enough to create a climate of serenity in families with children with special needs. Further efforts are necessary to meet the needs of parents of disabled children in terms of employment flexibility and social support, considering the needs not only of the disabled child but also the emotional and behavioral consequences in healthy siblings. Policy strategies should also support unaffected siblings through primary care and appropriate coping programs in order to create family-centered care aimed at having beneficial effects on all the family members.

Future research should focus on the family relationships through the administration of a different questionnaire for the child and for the parent. Moreover, a limitation of our study is the relatively small number of examined siblings in relation to the wide range of their ages. Therefore, we believe that other studies with larger samples in the future will be useful to further clarify the neuropsychological implications in healthy siblings of children with neurodevelopmental disorders.

The fraternal bond can be characterized as a positive experience and growth if, first of all, the possibility for children to freely experience their relationship is respected [34,35,36].

## Figures and Tables

**Table 1 medicina-56-00491-t001:** Strengths and Difficulties Questionnaire, Robert Goodman [27].

Scale	Questions	Scoring		
		Not True	Somewhat True	Certainly True
**Emotionality**	Often complains of headaches, stomach-aches or sickness	0	1	2
	Many worries or often seems worried	0	1	2
	Often unhappy, depressed or tearful	0	1	2
	Nervous or clingy in new situations, easily loses confidence	0	1	2
	Many fears, easily scared	0	1	2
**Behavioral Problems**	Often loses temper	0	1	2
	Generally well behaved, usually does what adults request	2	1	0
	Often fights with other children or bullies them	0	1	2
	Often lies or cheats	0	1	2
	Steals from home, school or elsewhere	0	1	2
**Hyperactivity and Inattention**	Restless, overactive, cannot stay still for long	0	1	2
	Constantly fidgeting or squirming	0	1	2
	Easily distracted, concentration wanders	0	1	2
	Thinks things out before acting	2	1	0
	Good attention span, sees work through to the end	2	1	0
**Relationship with Peers**	Rather solitary, prefers to play alone	0	1	2
	Has at least one good friend	2	1	0
	Generally liked by other children	2	1	0
	Picked on or bullied by other children	0	1	2
	Gets along better with adults than with other children	0	1	2
**Prosociality**	Considerate of other people’s feelings	0	1	2
	Shares readily with other children, for example toys, treats, pencils	0	1	2
	Helpful if someone is hurt, upset or feeling ill	0	1	2
	Kind to younger children	0	1	2
	Often offers to help others (parents, teachers, other children)	0	1	2

**Table 2 medicina-56-00491-t002:** Gender distribution among the three groups.

Siblings’ Disease		Frequency	Percentage
**ASD** **(Autism Spectrum Disorder)**	Male	29	54.7
	Female	24	45.3
	Total	53	100.0
**Down’s Syndrome**	Male	28	52.8
	Female	25	47.2
	Total	53	100.0
**Control Group**	Male	24	45.3
	Female	29	54.7
	Total	53	100.0

**Table 3 medicina-56-00491-t003:** Number of siblings in the three groups.

Siblings’ Disease		Age	Number of Children
**ASD**			53
	Mean	5.49	2.34
	Median	5.00	2.00
	Mode	3	2
	Standard Deviation	1957	0.783
**Down’s Syndrome**			53
	Mean	5.94	2.40
	Median	6	2
	Mode	5	2
	Standard Deviation	1985	0.716
**Control Group**			53
	Mean	5.89	2.26
	Median	6.00	2.00
	Mode	5	2
	Standard Deviation	1987	0.486

**Table 4 medicina-56-00491-t004:** Emotional difficulties subscale results.

Siblings’ Disease		No Discomfort	Subclinical Discomfort	Clinical Discomfort	Total
**ASD**	N.	26	12	15	53
	% *Siblings’ Disease*	49.1	22.6	28.3	100.0
	% Emotional Discomfort	27.1	29.3	71.4	33.5
**Down’s Syndrome**	N.	30	16	6	52
	% *Siblings’ Disease*	57.7	30.8	11.5	100.0
	% Emotional Discomfort	31.3	39.0	28.6	32.9
**Control Group**	N.	40	13	0	53
	% *Siblings’ Disease*	75.5	24.5	0.0	100.0
	% Emotional Discomfort	41.7	31.7	0.0	33.5
**Total**	N.	96	41	21	158
	% *Siblings’ Disease*	60.8	25.9	13.3	100.0
	% Emotional Discomfort	100.0	100.0	100.0	100.0

**Table 5 medicina-56-00491-t005:** Behavioral problems subscale results.

Siblings’ Disease		No Discomfort	Subclinical Discomfort	Clinical Discomfort	Total
**ASD**	N.	42	4	7	53
	% *Siblings’ Disease*	79.2	7.5	13.2	100.0
	% Emotional Discomfort	37.2	16.7	31.8	33.5
**Down’s Syndrome**	N.	35	8	10	53
	% *Siblings’ Disease*	66.0	15.1	18.9	100.0
	% Emotional Discomfort	31.0	33.3	45.5	33.3
**Control Group**	N.	36	12	5	53
	% *Siblings’ Disease*	67.9	22.6	9.4	100.0
	% Emotional Discomfort	31.9	50.0	22.7	33.3
**Total**	N.	113	24	22	159
	% *Siblings’ Disease*	71.1	15.1	13.8	100.0
	% Emotional Discomfort	100.0	100.0	100.0	100.0

**Table 6 medicina-56-00491-t006:** Hyperactivity and inattention subscale results.

Siblings’ Disease		No Discomfort	Subclinical Discomfort	Clinical Discomfort	Total
**ASD**	N.	36	11	6	53
	% *Siblings’ Disease*	67.9	20.8	11.3	100.0
	% Emotional Discomfort	30.3	44.0	40.0	33.3
**Down’s Syndrome**	N.	37	8	8	53
	% *Siblings’ Disease*	69.8	15.1	15.1	100.0
	% Emotional Discomfort	31.1	32.0	53.3	33.3
**Control Group**	N.	46	6	1	53
	% *Siblings’ Disease*	86.8	11.3	1.9	100.0
	% Emotional Discomfort	38.7	24.0	6.7	33.3
**Total**	N.	119	25	15	159
	% *Siblings’ Disease*	74.8	15.7	9.4	100.0
	% Emotional Discomfort	100.0	100.0	100.0	100.0

**Table 7 medicina-56-00491-t007:** Relationship with peers subscale results.

Siblings’ Disease		No Discomfort	Subclinical Discomfort	Clinical Discomfort	Total
**ASD**	N.	28	11	14	53
	% *Siblings’ Disease*	52.8	20.8	26.4	100.0
	% Emotional Discomfort	31.1	29.7	43.8	33.3
**Down’s Syndrome**	N.	24	15	14	53
	% *Siblings’ Disease*	45.3	28.3	26.4	100.0
	% Emotional Discomfort	26.7	40.5	43.8	33.3
**Control Group**	N.	38	11	4	53
	% *Siblings’ Disease*	71.7	20.8	7.5	100.0
	% Emotional Discomfort	42.2	29.7	12.5	33.3
**Total**	N.	90	37	32	159
	% *Siblings’ Disease*	56.6	23.3	20.1	100.0
	% Emotional Discomfort	100.0	100.0	100.0	100.0

**Table 8 medicina-56-00491-t008:** Pro-social behaviour subscale results.

Siblings’ Disease		No Discomfort	Subclinical Discomfort	Clinical Discomfort	Total
**ASD**	N.	41	8	4	53
	% *Siblings’ Disease*	77.4	15.1	7.5	100.0
	% Emotional Discomfort	32.8	42.1	26.7	33.3
**Down’s Syndrome**	N.	39	7	7	53
	% *Siblings’ Disease*	73.6	13.2	13.2	100.0
	% Emotional Discomfort	31.2	36.8	46.7	33.3
**Control Group**	N.	45	4	4	53
	% *Siblings’ Disease*	84.9	7.5	7.5	100.0
	% Emotional Discomfort	36.0	21.1	26.7	33.3
**Total**	N.	125	19	15	159
	% *Siblings’ Disease*	78.6	11.9	9.4	100.0
	% Emotional Discomfort	100.0	100.0	100.0	100.0

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
