# Peer review of "Emotional–Behavioral Disorders in Healthy Siblings of Children with Neurodevelopmental Disorders"

_medicina, 2020, doi:10.3390/medicina56100491_

Round 1

Reviewer 1 Report

Thank you for the opportunity to review this manuscript (medicina-916213) examining Emotional-behavioral disorders in healthy siblings of children with neurodevelopmental disorders. A particular strength of the manuscript is its potential to address an important gap in the literature by examining an understudied population. Moreover, the topic is explored as thoroughly within non-US based samples. As such, the findings have the potential to make a contribution to the literature and inform public health practice. The background and justification for the study is thoroughly discussed and engrained within the academic literature. There are some minor revisions that are necessary, as outlined below:
1. Abstract. The authors could add some lines about clinical and policy implications.

  1. Regarding similar studies conducted in the Italian context, in the Introduction, as well as in the Conclusions, the authors could cite other more recent researches: e.g., De Caroli, M. E., & Sagone, E. (2013). Siblings and disability: a study on social attitudes toward disabled brothers and sisters. Procedia - Social and Behavioral Sciences, 93, 1217–1223; Di Biasi, S., Trimarco, B., D’Ardia, C., Melogno, S., Meledandri, G., & Levi, G. (2015). Psychological adjustment, social responsiveness and parental distress in an Italian sample of siblings of children with high-functioning autism spectrum disorder. Journal of Child and Family Studies, 25(3), 883–890; Laghi, F., Lonigro, A., Pallini, S., Bechini, A., Gradilone, A., Marziano, G., et al. (2018). Sibling relationships and family functioning in siblings of early adolescents, adolescents and young adults with autism spectrum disorder. Journal of Child and Family Studies, 27, 793–801; Prino, L. E., Scigala, D., Fabris, M. A., & Longobardi, C. (2019). The moderating role of gender in siblings of adults with intellectual disabilities. Interpersona. An International Journal on Personal Relationships, 13(1), 1–13; Sommantico, M., Parrello, S., De Rosa, B. (2020). Sibling relationships, disability, chronic, and mental illness: Development of the Siblings’ Experience Quality Scale (SEQS). Journal of Developmental and Physical Disabilities, Fisrt Online 28 January. doi: 10.1007/s10882-020-09730-4; Sommantico, M., Parrello, S., De Rosa, B. (2020). Adult siblings of people with and without intellectual and developmental disabilities: Sibling relationship attitudes and psychosocial outcomes. Research in Developmental Disabilities, 99, 103594. doi: 10.1016/j.ridd.2020.103594;
  2. Materials and Methods: the authors could add a section regarding the analysis plan.
  3. Results:
  4. a) Why the authors only analyzed the single subscales for each group, and not also the entire scale?
  5. b) Why the authors only utilized the SDQ-ITA questionnaire, without any other instruments as outcome variables?
  6. Discussion and Conclusions: The authors could deeply discuss clinical and policy implications of their findings

Author Response

Dear Reviewer,

I would like to thank you for your valued comments and suggestions to the article. As you requested, we made all the necessary changes in our manuscript to address the reviewers’ concerns and we detailed below how the points raised by the referees have been accommodated. The main changes are written in red in the text of the manuscript. From the changes made in the revised manuscript and responses provided below, I hope you are convinced that we have adequately addressed the reviewer’s concerns and made the paper better. If there are any further questions, please feel free to let me know.

Reviewer 1

Thank you for the opportunity to review this manuscript (medicina-916213) examining Emotional-behavioral disorders in healthy siblings of children with neurodevelopmental disorders. A particular strength of the manuscript is its potential to address an important gap in the literature by examining an understudied population. Moreover, the topic is explored as thoroughly within non-US based samples. As such, the findings have the potential to make a contribution to the literature and inform public health practice. The background and justification for the study is thoroughly discussed and engrained within the academic literature. There are some minor revisions that are necessary, as outlined below:

  1. The authors could add some lines about clinical and policy implications.

Thanks for the suggestion. Clinical and policy implications have been mentioned in the abstract (lines 35-36) and in the conclusion (lines 345-352).

  1. Regarding similar studies conducted in the Italian context, in the Introduction, as well as in the Conclusions, the authors could cite other more recent researches: e.g.,

De Caroli, M. E., & Sagone, E. (2013). Siblings and disability: a study on social attitudes toward disabled brothers and sisters. Procedia - Social and Behavioral Sciences, 93, 1217–1223;

Di Biasi, S., Trimarco, B., D’Ardia, C., Melogno, S., Meledandri, G., & Levi, G. (2015). Psychological adjustment, social responsiveness and parental distress in an Italian sample of siblings of children with high-functioning autism spectrum disorder. Journal of Child and Family Studies, 25(3), 883–890;

Laghi, F., Lonigro, A., Pallini, S., Bechini, A., Gradilone, A., Marziano, G., et al. (2018). Sibling relationships and family functioning in siblings of early adolescents, adolescents and young adults with autism spectrum disorder. Journal of Child and Family Studies, 27, 793–801;

Prino, L. E., Scigala, D., Fabris, M. A., & Longobardi, C. (2019). The moderating role of gender in siblings of adults with intellectual disabilities. Interpersona. An International Journal on Personal Relationships, 13(1), 1–13;

Sommantico, M., Parrello, S., De Rosa, B. (2020). Sibling relationships, disability, chronic, and mental illness: Development of the Siblings’ Experience Quality Scale (SEQS). Journal of Developmental and Physical Disabilities, Fisrt Online 28 January. doi: 10.1007/s10882-020-09730-4;

Sommantico, M., Parrello, S., De Rosa, B. (2020). Adult siblings of people with and without intellectual and developmental disabilities: Sibling relationship attitudes and psychosocial outcomes. Research in Developmental Disabilities, 99, 103594. doi: 10.1016/j.ridd.2020.103594;

Thanks for the suggestions. We added and commented the suggested articles (please see lines 83-86, 97-106, 313-314)

  1. Materials and Methods: the authors could add a section regarding the analysis plan.

We have better explained how the data analysis was performed (please see lines 167-168).

  1. a) Why the authors only analyzed the single subscales for each group, and not also the entire scale?

Thanks for the suggestion. We added the overall difficulties scores for all groups (please see lines 200-202).

  1. b) Why the authors only utilized the SDQ-ITA questionnaire, without any other instruments as outcome variables?

The SDQ-ITA questionnaire was chosen because it is provided with good psychometric features and it is short, rapid and easy to administer. Moreover, compared to other instruments (CBCL, CDI e Connors scales) the SDQ-ITA questionnaire is the only one capable of evaluating children from 3 years old.

  1. Discussion and Conclusions: The authors could deeply discuss clinical and policy implications of their findings

Thanks for the suggestion. Clinical and policy implications have been mentioned in the abstract (lines 35-36) and in the conclusion (lines 345-352).

Reviewer 2 Report

The article refers to an important research problem that is relatively rarely discussed in the literature on the subject. The authors assessed the emotional-behavioral disorders occurring in siblings of children with neurodevelopmental disorders. The basis for the diagnosis were the results of the Strengths and Difficulties Questionnaire completed by the parents of these children. In the Introduction, the authors rightly emphasize that the presence of a disabled child in the family may have both negative and positive consequences. They also mention that the concept of their research is based on similar studies previously conducted in Iran (2018). The methodological part of the work is generally correctly prepared. However, the authors should comment on the relatively wide age range of the studied children, i.e. from 3 to 9 years of age. On the one hand, we have preschool children and, on the other, school-age children. It is known that they differ in their behavior, and some symptoms of a behavioral disorder may be due to their development cycle rather than to the presence of a disabled sibling. This problem needs to be clarified.
The discussion of the results contains significant shortcomings. There are no comparisons with the results of similar studies. For example, on page 10 (line 265) the authors state 'This is in line with other previous studies', but it is not known what author they are citing. Moreover, they should compare their results with the research carried out in Iran, as they are mentioned in the Introduction section.
At the end of the work, the authors should add a comment about the limitations related to the obtained results. It is necessary due to the relatively small number of examined children in relation to the wide range of their ages.
Additionally, authors should avoid phrases that place disabled children in a negative light. For example, instead of 'children with autism and Down's syndrome could cause greater psychological stress to their families and to their siblings' (line 280-281) there should be 'presence of children with autism and Down's syndrome could cause greater psychological stress to their families and to their siblings'. It is obvious that disabled children do not cause any stress in the family consciously and actively.
The bibliography is prepared with carelessly. The name of the journal has been omitted in many items (e.g. no. 3, 7, 8, 14, 15, 16, 17, 19). Authors should also remember that in English journal names are written with a capital letter (e.g. no. 11).
Technical note. In tables 4-7, there is no need to put the % symbol next to each number if that symbol has been explained in the adjacent column. Moreover, instead of "Counting" there should be the symbol 'N' implemented.

Author Response

Dear Reviewer,

I would like to thank you for your valued comments and suggestions to the article. As you requested, we made all the necessary changes in our manuscript to address the reviewers’ concerns and we detailed below how the points raised by the referees have been accommodated. The main changes are written in red in the text of the manuscript. From the changes made in the revised manuscript and responses provided below, I hope you are convinced that we have adequately addressed the reviewer’s concerns and made the paper better. If there are any further questions, please feel free to let me know.

The article refers to an important research problem that is relatively rarely discussed in the literature on the subject. The authors assessed the emotional-behavioral disorders occurring in siblings of children with neurodevelopmental disorders. The basis for the diagnosis were the results of the Strengths and Difficulties Questionnaire completed by the parents of these children. In the Introduction, the authors rightly emphasize that the presence of a disabled child in the family may have both negative and positive consequences. They also mention that the concept of their research is based on similar studies previously conducted in Iran (2018).

The methodological part of the work is generally correctly prepared. However, the authors should comment on the relatively wide age range of the studied children, i.e. from 3 to 9 years of age. On the one hand, we have preschool children and, on the other, school-age children. It is known that they differ in their behavior, and some symptoms of a behavioral disorder may be due to their development cycle rather than to the presence of a disabled sibling. This problem needs to be clarified.

We thank Reviewer for the valued suggestion. Analysis did not reveal a statistically significant difference related to the age of the subjects who submitted the questionnaire. However, thanks to your suggestion we clarified (in the discussion section) the choice of the age range and we underlined the hope that further studies in the future could investigate separately preschool children and school-age children (lines 298-305)

The discussion of the results contains significant shortcomings. There are no comparisons with the results of similar studies. For example, on page 10 (line 265) the authors state 'This is in line with other previous studies', but it is not known what author they are citing.

The correct reference has been added.

Moreover, they should compare their results with the research carried out in Iran, as they are mentioned in the Introduction section.

Thanks for the suggestion. We made clear now the differences between our results and Pourbagheri et al. results (please see the lines: 276-281)

At the end of the work, the authors should add a comment about the limitations related to the obtained results. It is necessary due to the relatively small number of examined children in relation to the wide range of their ages.

We inserted as suggested this limitation in the end of our article, thanks for the suggestion (see lines 354-357).

Additionally, authors should avoid phrases that place disabled children in a negative light. For example, instead of 'children with autism and Down's syndrome could cause greater psychological stress to their families and to their siblings' (line 280-281) there should be 'presence of children with autism and Down's syndrome could cause greater psychological stress to their families and to their siblings'. It is obvious that disabled children do not cause any stress in the family consciously and actively.

I appreciated a lot your suggestion and I corrected it.

The bibliography is prepared with carelessly. The name of the journal has been omitted in many items (e.g. no. 3, 7, 8, 14, 15, 16, 17, 19). Authors should also remember that in English journal names are written with a capital letter (e.g. no. 11).

Thanks for the suggestion. All references have been reformatted using the bibliography software EndNote and the MDPI style file

Technical note. In tables 4-7, there is no need to put the % symbol next to each number if that symbol has been explained in the adjacent column. Moreover, instead of "Counting" there should be the symbol 'N' implemented.

Thanks for your suggestions. I made the necessary adjustments as required.